# *GLI2* and *FLNB* Define Periocular Morphoeic Basal Cell Carcinoma

**DOI:** 10.3390/ijms262311377

**Published:** 2025-11-25

**Authors:** John C. Bladen, Jun Wang, Mariya Moosajee, Muhammad Rahman, Ajanthah Sangaralingam, Vijay K. Gogna, Claude Chelala, Edel A. O’Toole, Michael P. Philpott

**Affiliations:** 1Centre for Cell Biology and Cutaneous Research, Blizard Institute, London E1 2AT, UK; m.m.rahman@qmul.ac.uk (M.R.);; 2Department of Oculofacial & Orbital Surgery, Kings College Hospital, London SE5 9RS, UK; 3Centre for Molecular Oncology, Barts Cancer Institute, Queen Mary University of London, London EC1M 6AU, UK; 4Department of Ocular Biology and Therapeutics, UCL Institute of Ophthalmology, London EC1V 9EL, UK; m.moosajee@ucl.ac.uk; 5Centre for Cancer Biomarkers and Biotherapeutics, Barts Cancer Institute, Queen Mary University of London, London EC1M 6AU, UK

**Keywords:** morphoeic, basal cell carcinoma, *GLI2*, *FLNB*, hedgehog signalling, ultraviolet

## Abstract

Morphoeic basal cell carcinoma (mBCC) has a higher risk of local recurrence than the more indolent nodular (nodBCC) subtype. Little is known about the genetic and molecular makeup of mBCC that determines its invasive behaviour: a comparison of mBCC with nodBCC was carried out. Whole-exome sequencing (WES) of 20 BCC tumours (10 eyelid morphoeic and 10 nodular) underwent driver gene detection using OncodriveFM and MutSigCV, followed by a randomisation analysis procedure. Samples underwent RNA sequencing, gene-set enrichment analysis and candidates verified by RT-PCR. *PTCH1*, *FLNB,* and double-knockdown human keratinocyte models were used to validate phenotype and gene expression. Hedgehog pathway analysis of 20 additional BCCs underwent immunostaining verification. Our analysis revealed FLNB as a potential driver with a mutational cluster in *FLNB* Filamin domain 24 and a 4-fold reduction in expression compared to normal eyelids in mBCC only. *FLNB* knockdown demonstrated an mBCC phenotype. Aberrant Gli2 dominant hedgehog (Hh) signalling was seen in mBCC on three molecular levels: mutational significance, transcriptome profile, and protein expression. Gli2-dominant Hh overexpression was seen in the tumour plus stroma of eyelid morphoeic but not nodular BCC. *FLNB* is a potential tumour suppressor, with its loss producing a morphoeic phenotype in vitro.

## 1. Introduction

Basal cell carcinoma (BCC) is the most common cancer, with 60% affecting the face [1,2]. BCCs situated within the H zone of the face, including the eyelid region, have a higher risk of recurrence, behave more aggressively, and pose a risk to sight [3,4]. Increased risk of recurrence is a histological diagnosis of morphoeic BCC (mBCC) [5,6]. Little is known about what makes mBCC more locally aggressive, but it comprises small spikey strands of cancer cells with surrounding inflammatory fibrosis within the stroma, which requires margin-controlled excision to ensure complete histological resection [7,8]. There are no studies of the molecular behaviour of eyelid BCC, partly due to its lower incidence (4.5 per 100,000) compared to BCC on the rest of the body (200 per 100,000) but also due to its precarious location and risk of morbidity to obtain a good tissue sample above what is needed for diagnosis; nevertheless, the incidence of BCC is increasing in all locations [9,10,11]. Furthermore, eyelid contains a tarsal plate below the skin, which is a unique microenvironment with layers of collagen surrounded by fibroblasts, which may predispose the eyelid region to unique cancer drivers [12,13].

The genetics of BCC have been described by the analysis of basal cell naevus syndrome (Gorlin syndrome), an autosomal-dominant, heterozygous, germline deletion of the *PTCH 1* gene [14,15,16,17]. Homozygosity in *PTCH1* (a second *PTCH1* deletion, for example, from UV light damage) in BCNS is required for cancer formation, but inherited BCC accounts for 1–2% of BCC; the vast majority of BCC is sporadic [18]. The head is a common location supporting the UV theory; however, up to a third occurs in intermittent- or no-UV-light areas [19]. Sporadic BCCs show heterozygous mutations in *PTCH1* and other hedgehog (Hh) signalling genes (e.g., SMO) but is not universal [20,21,22]. Genomic analysis of 293 BCC revealed seven significantly mutated genes (*PTCH1, TP53, SMO, MYCN, PTPN14, RPL22* and *PPIAL4G*) and dysregulated pathways such as TGF-β plus TP53, but there was a bias towards the nodular subtype, with only eight mBCC undergoing whole-exome sequencing (WES) and not from the eyelid region [23,24].

Cancer behaviour traverses tumour borders delineated on microscopy; the surrounding apparent non-neoplastic tissue has been shown to play a key role in local spread, with stromal Hh signalling supporting tumorigenesis [25,26]. We have shown that stromal modulation promotes mBCC [27]. However, comparisons of genetic and transcriptomic changes in the stroma of nodular and morphoeic BCCs have not been made.

We report potential driver mutations and pathways within eyelid mBCC in comparison with the indolent nodular subtype using a multiomics approach. We validated a key mBCC driver gene, *FLNB,* using a *PTCH* null BCC keratinocyte migration model and in a 3D model of cell invasion [28], showing knockdown of *FLNB* drives a morphoeic migratory and invasive phenotype. Finally, we show upregulated Gli2 dominant Hh signalling is characteristic of both the tumour and the stroma in mBCC but not nBCC.

## 2. Results

### 2.1. Distinct Driver Genes Define Morphoeic and Nodular BCC

We analysed somatic mutations in 10 eyelid mBCCs using OncodriveFM and revealed 16 significantly mutated genes (*q* < 0.1), including two known drivers, *PTCH1* and *SMARCA4*, and genes connected to a driver, such as *EPHA3, FLNB,* and *FLNC* (Figure 1A, Appendix A). MutSigCV identified five driver genes (*p* < 0.02, Appendix A) including *PTCH1*, which was the single common driver across both algorithms (Figure 1A).

In contrast, in eyelid nodBCC OncodriveFM highlighted 15 mutated genes (*q* < 0.1; Appendix A, available online), and MutSigCV indicated 11 genes (*p* < 0.02; Appendix A, available online). The shared genes across the two algorithms revealed six drivers, *PTCH1*, *TP53*, *FADS1, PPM1D, TAF4B,* and *TRIP4* (Figure 1B).

Randomisation tests further highlighted significantly mutated genes specific to an eyelid BCC subtype (Figure 1C). OncodriveFM mBCC-specific genes included *FLNB* and *HECTD4,* and nodBCC-specific genes were *ZEB1* and *TP53* (Figure 1C, Appendix A). *FLNB* demonstrated a mutational cluster in Filamin domain 24, including a hotspot P2586S in three mBCCs and V2602L in one mBCC (Figure 2A). For *HECTD4*, mutations were clustered near the N-terminal (Figure 2B). RNA-seq data showed a trend towards downregulation of *FLNB* and *HECTD4* in mBCC compared to normal eyelids. Moreover, a 4-fold reduction in *FLNB* and *HECTD4* was demonstrated using qRT-PCR in mBCC compared to normal eyelid tissue, suggesting downregulation in mBCC (Appendix A). Copy loss in *FLNB* in 3 out of 20 mBCC is also supportive of this (Figure 1C). The MutSigCV mBCC-specific gene included *CCDC108*, and nodBCC-specific genes included *GABRA6*, *OR52J3*, *TRIM39,* and *ADAM29* (Figure 1C; Appendix A).

### 2.2. Copy Number Aberration in BCC

The percentage of the genome affected by CNAs ranged from 0 to 28% (mean 8.6%) in eyelid mBCC and 0.2 to 19.5% (mean 9.2%) in eyelid nodBCC, with the overall profile being similar in both subtypes (Appendix A). Chromosome 9q loss or copy neutral loss-of-heterozygosity (cn-LOH) was the most frequent CNA in BCC (eight mBCCs and seven nodBCCs).

### 2.3. Mutational and Transcriptome Expression Pathway Analysis

Four pathways showed significant dysregulation in mBCC at both the DNA and RNA levels (*q* < 0.05 in WES using OncodriveFM and in RNA-seq using GSEA). These included (i) the hedgehog (Hh) signalling pathway, (ii) the BCC pathway, (iii) natural killer cell-mediated cytotoxicity, and (iv) the Fc epsilon RI signalling pathway (the latter meeting a threshold of *q* = 0.07 in both datasets) (Figure 3A, Appendix A). In nodBCC, three pathways were dysregulated across both mutational and transcriptomic data: (i) the BCC pathway, (ii) PPAR signalling, and (iii) oxidative phosphorylation (Figure 3B, Appendix A). Notably, in nodBCC, the Hh, TGF-beta, and p53 signalling pathways displayed strong mutational significance; however, this was not mirrored at the transcriptomic level (Figure 3B).

Analysis of WES pathway data comparing mBCC and nodBCC identified Wnt signalling as a newly significant pathway shared between the two subtypes. Additional pathways common to both included (i) the BCC pathway, (ii) Hh signalling, (iii) pathways in cancer, and (iv) the cell cycle (Figure 3C). Natural killer cell-mediated cytotoxicity and Fc Epsilon RI signalling were significantly impacted by somatic mutations in mBCC but not in nodBCC. When RNA-seq GSEA profiles from nodBCC and mBCC were compared, several immune-related pathways emerged as shared, including natural killer cell-mediated cytotoxicity. However, marked upregulation of Hh signalling was detected exclusively in the mBCC transcriptome (Figure 3D).

### 2.4. GLI2 Overexpression Characterises Eyelid Morphoeic BCC Phenotype

qRT-PCR data revealed 16-fold overexpression of *GLI2* in eyelid mBCC compared to normal eyelid (*p* < 0.01) but not in eyelid nodBCC, and an overexpression was seen in the surrounding mBCC stroma (*p* < 0.05, Figure 4A). Protein expression demonstrated upregulation of the Hh pathway in mBCC tumours including GLI1 and GLI2, whereas nodBCC had a normal level of expression of the Hh pathway, not above the levels of physiological activation (Figure 4B, Appendix A). GLI2 and other Hh proteins were also overexpressed in mBCC stroma (Figure 4C).

### 2.5. FLNB and PTCH1-FLNB Knockdown Mimics Morphoeic BCC Phenotype

Loss-of-function mutation in *PTCH1* is common to both mBCC and nodBCC (Figure 1A,B). To investigate subtype-specific driver genes and the distinctive expression of the Hh pathway, we used a model of *PTCH1* knockdown in human NEB-1 keratinocytes (NEB1-shPTCH1), which resembles nodular but not mBCC [28] and induced a compact keratinocyte morphology (Figure 5A) and reduced migration in a scratch assay (Figure 5B). In *PTCH1* knockdown cells there was a significant upregulation of Hh-pathway genes (*GLI1* and *SMO*), *LOXL2*, *NFIB*, the morphoeic driver *HECTD4* (downregulated in in vivo mBCC) and downregulation in *GLI2* (upregulated in in vivo mBCC) and no change in *FLNB* (downregulated in in vivo mBCC) (Figure 6), mimicking a nodular genotype.

Two in vitro mBCC cell lines were constructed with a double knockdown (*PTCH1* and *FLNB),* attempting to create a more realistic model of mBCC. *FLNB* was chosen as it was identified in every step in the in vivo data and predicted function (SIFT and PolyPhen), making it a good candidate to investigate further. *FLNB* knockdown in NEB1 cells resulted in more disorganised and less-cell-dense NEB1 colonies (Figure 5A) and greater migratory potential post-scratch compared to control cells (Figure 5B). Furthermore, as previously shown, *PTCH* knockdown inhibits cell migration [28]. Suppression of *FLNB* in *PTCH* knockdown cells enhances migration despite *PTCH1* suppression (NEB1-shPTCH1-shFLNB; Figure 5B). Three-dimensional models of invasion into collagen gels containing dermal fibroblasts demonstrate invasive behaviour of *FLNB*-knockdown cells with the attraction of fibroblasts, a hallmark of morphoeic BCC (Figure 6). In vivo mBCC shows disorganisation, enhanced migration, and attraction of fibroblasts (Appendix A). Therefore, *FLNB* knockdown showed colony disorganisation, enhanced migration in the scratch assay, and invasion in the 3D organotypics with fibroblast attraction, highlighting its role in developing a ‘morphoeic-like’ phenotype.

Subsequently, a comparison of with our human eyelid RNA-sequencing data for nodular and morphoeic BCC was made with our three in vitro cell models: *PTCH, FLNB,* and double *PTCH/FLNB* knockdown (Figure 7). Consistent with our human mBCC RNA-seq data, *FLNB* and *PTCH/FLNB* double knockdown had little impact on nBCC genes apart from upregulation of *TGF-β2*. *FLNB* knockdown resulted in upregulation of MYCN compared to *PTCH1* knockdown only and further MYCN upregulation by double-*PTCH/FLNB* knockdown. *EPHA3* and *HECTD4* were attenuated by both FLNB and FLNB/PTCH knockdown. *PLAT* and *TRIM22* were upregulated by a greater extent in the double-*PTCH/FLNB* knockdown (mBCC phenotype) compared to PTCH1 knockdown (nodBCC phenotype) alone.

### 2.6. Morphoeic Stromal Microenvironment Contains Potential Driver Genes and Abnormal Hedgehog Signalling

WES of one eyelid mBCC tumour–stroma pair identified 150 shared nonsynonymous mutations—representing 76.5% of all stromal mutations—including mutations in *ATR*, *EPHA3* and *GLI3* (Appendix A). In the mBCC stroma, 37 genes were differentially expressed, with *GSTM1* showing the highest level of upregulation (Appendix A). GSEA indicated upregulation of immune-related pathways in mBCC stroma relative to normal eyelid stroma (Appendix A). Additionally, *GLI1* and *GLI2* RNA (*p* < 0.01) and protein expression were significantly elevated in mBCC stroma but not observed in nodBCC stroma (Figure 4C; Appendix A). Nevertheless, being only one sample, this is only a preliminary finding.

## 3. Discussion

This study comprises the largest genetic analysis and comprehensive multiomics molecular assessment of eyelid BCC within the high-risk H region of the face and specifically the locally aggressive mBCC. As far as we are aware, this study also represents the largest genetic analysis of morphoeic BCC, which, while less common than nBCC, has a greater morbidity [5,6,9,29] and is poorly understood both in terms of genetics and cell biology.

*PTCH1* alone does not result in a morphoeic phenotype being common to both mBCC and nodBCC; however, loss-of-function (as a potential tumour-suppressor gene) drivers *FLNB* and *HECTD4* were identified as being morphoeic-specific. Loss of *FLNB* inhibits vascular permeability and, in ovarian cancer cells, promotes MMP9-mediated tumour invasion and VEGFA-mediated angiogenesis plus induces tumour growth [30,31,32]. A functionally damaging mutation hotspot P2586S was identified at Filamin 24 occurring in three mBCC cases, potentially impairing its interaction with FLNA2. *HECTD4* is a ubiquitin ligase family member that is involved in a range of cellular functions; intronic SNV in this gene (rs11066280) has been associated with a poorer outcome in oesophageal squamous cell carcinoma [33]. Furthermore, it has been shown to have a tumour- and metastasis-suppressor role [34].

We have previously shown that *PTCH1* knockdown gives rise to compact colonies similar to nBCC [28]. Functional analysis using 2D and 3D *FLNB* and *FLNB-PTCH1* knockdown models demonstrated a ‘morphoeic-like’ phenotype with less compact colonies, more migratory and invasive phenotype, and attraction of fibroblasts, which is a specific hallmark required for the diagnosis of morphoeic BCC in vivo. *LOXL* and *NFIB,* both characteristic of nBCC, were significantly upregulated by *PTCH1* knockdown. *TRIP4*, which is associated with malignant melanoma, was repressed by *PTCH1* but to a lesser extent than *PTCH1/FLNB* double knockdown, perhaps reflecting the more aggressive nature of mBCC [35]. Genes associated with morphoeic eyelid BCC, *TRIM22* and *PLAT*, were significantly upregulated in *PTCH1* knockdown but further increased by *PTCH1/FLNB* double knockdown. *EPHA3* is an inhibitor of cell migration and invasion, with missense mutation reported in locally advanced recurring BCC [36]. *EPHA3* was significantly u-regulated by *PTCH1* knockdown, and this was attenuated by double-*PTCH/FLNB* knockdown, highlighting nodular characteristics of *PTCH1* knockdown; *MYCN* was significantly downregulated by *PTCH1* knockdown: a higher prevalence of strong *MYCN* expression has been reported in BCC with an infiltrative growth pattern, further supporting the hypothesis that different molecular pathways contribute to the formation of BCC subtypes and that *PTCH1* knockdown may be more closely associated with nodular than mBCC [37].

Six nodBCC-specific genes, including *TP53* and *ZEB1,* were identified as integral factors in nodular BCC development. *ZEB1* plays a fundamental role in tumour progression and confers a poorer clinical outcome in cancer; nevertheless, PI3K-targeted therapy can suppress the metastatic drive by *ZEB1*, representing a novel treatment target in nodBCC [38]. Bonilla et al. identified seven driver genes in their analysis of BCCs containing mixed histological subtypes, and we observed a similar mutation frequency in our 20 nBCC cases: *PTCH1* (75%), *TP53* (45%), *SMO* (15%), *MYCN* (25%), *PTPN14* (10%), *RPL22* (20%), and *PPIAL4G* (10%). Although transcriptome pathway analysis by Bonilla et al. revealed TP53 and TGF-β signalling as significantly upregulated pathways in BCC, this was revealed by mutational significance only in our data and did not reflect the genotype of eyelid mBCC [23].

WES, GSEA, and protein validation of mBCC demonstrate *GLI2*-predominant aberrant Hh expression. Furthermore, protein validation also demonstrated GLI2-dominated Hh expression in the stroma of morphoeic BCC but not in nodBCC. Expression of *Gli2* in mouse epidermis is sufficient to induce BCC-like tumours [39]; in human keratinocytes, it induces a number of genes involved in G1–S phase and G2–M phase progression [40] and induces genomic instability by inhibiting apoptosis [41]. Moreover, it promotes keratinocyte invasion in 3D histotypic models, which appears to be mediated via downregulation of E-cadherin and direct upregulation of Wnt signalling, a possible core tumourigenic BCC pathway [42,43,44,45]. Upregulation of *GLI2* in morphoeic BCC adds further support to the suggestion that *GLI2* rather than *GLI1* may encode the critical factor involved in executing oncogenic hedgehog signalling in response to constitutive pathway activation [46]. Thus, the lack of *GLI2* expression in our *PTCH1*-knockdown cellular model may explain its nodular phenotype.

There is likely a signal-induced regulatory mechanism that contributes to the cancer activity of *GLI2* that is not part of canonical hedgehog signalling. *PDGFB* is overexpressed in morphoeic tumours compared to nodular tumours by four folds (*p* = 0.01), and the PDGFB-GLI2 axis is known to modulate cancer stem cell properties; thus, this axis may be more important in determining the *GLI2* dominance seen in morphoeic BCC, although this requires future investigation to support this hypothesis [47].

Wnt and Hh signalling are shared pathways in BCC. The former has been shown in a transcriptome analysis of sporadic nodBCC [48], and we highlight its importance in the morphoeic subtype too, suggesting it represents an important core tumorigenic pathway common to both BCC subtypes.

Activation of stromal Hh pathway could occur as a result of local paracrine signals from the morphoeic tumour or be related to the mutated stroma itself including *GLI3* [49]. Differential stromal gene expression revealed *GSTM1*, a detoxification enzyme associated with BCC, to be expressed in mBCC but not nodBCC [50,51]. Chronic inflammation in the microenvironment and evasion of the immune system are emerging characteristics of tumourigenesis [52,53,54]. The former is present in mBCC, and the latter shown to be in both. Natural killer cell-mediated cytotoxicity (NK) was a significantly mutated pathway in mBCC and was significantly downregulated in both subtypes, although the exact role of NK in BCC is unclear [55]. Nonetheless, cancer-induced local immune evasion/immunosuppression is known to occur [56,57,58].

Limitations of the study include the small sample size with only 20 BCCs, especially for the stroma WES, with only once sample. The NEB1 keratinocyte microenvironment is not the same as the eyelid microenvironment so direct interpretation is limited.

To conclude, the driver mutational profile is different in mBCC, with loss of function of potential tumour suppressors *FLNB* and *HECTD4* compared to its indolent nodular counterpart. Despite loss-of-function *PTCH1* being common to both subtypes, the differing expressions of Hh, specifically *GLI2*, may contribute to their markedly differing behaviour, especially within the surrounding morphoeic stroma. Moreover, tumourigenic behaviour will probably be enhanced by the chronic inflammation seen in mBCC coupled with the loss of NK cytotoxicity. Recurrence seen in mBCC may therefore be due to stromal mutational burden or cancer cells undergoing epithelial–mesenchymal transition intermixed within the stromal environment; however, future studies are required to determine this.

## 4. Materials and Methods

### 4.1. Patient and Tumour Samples

A prospective collection of BCC tumours from 20 patients (10 nodular and 10 morphoeic) underwent whole-exome sequencing (WES) (human investigations were performed after approval by national Research Ethics Committee 119204/426396/1/202). Ten eyelid mBCCs and ten eyelids nodBCC were recruited (Appendix A). Average age for 10 patients with mBCC was 72.3 years and 72.4 for the 10 patients with nodBCC. Morphoeic and nodular histological subtypes were pure in definition, with mixed phenotypes excluded [59]. Specifically, morphoeic is defined as a high-risk subtype with infiltrating irregular groups of cells accompanied by stromal fibrosis—a histological hallmark of its aggressive behaviour (Appendix A). Laser capture microdissection (LCM) was used for accurate isolation of tumour, stroma, and normal tissue for DNA and RNA extraction, a technique that has been perfected in our lab; expression of cytokeratin plus TP63 confirms the quality of the LCM (Appendix A) [60]. Stroma is defined as non-neoplastic tissue adjacent to tumour. An additional 20 eyelid BCC (10 morphoeic and 10 nodular) samples underwent immunohistochemistry. Informed consent was obtained each patient including for photographic publication.

### 4.2. Whole-Exome Sequencing (WES)

WES was performed on DNA isolated by LCM from fresh tumour tissue and paired, matched blood controls using a Qiagen QIAamp Micro kit and a Qiagen DNeasy Blood and Tissue kit, respectively. Gene encoding regions covering 1.5% of the human genome were sequenced. At least 1.5 µg of DNA, quantified using a Qubit 2.0 Fluorometer (Life Technologies, Carlsbad, CA, USA), was used for library preparation. Libraries were prepared with an Agilent SureSelect (51 Mb) version 5 capture kit, and WES was conducted on an Illumina HiSeq 2000 platform using 100 bp paired-end reads to achieve a minimum mean coverage of 100X, performed by Oxford Gene Technology (OGT). Output FASTQ files were processed and aligned to the hg19/GRCh37 reference genome, and somatic variants and CNAs from the tumour–normal pair were identified and annotated using our previously described bioinformatics pipeline [61,62]. Functional predictions for non-synonymous variants were generated using PolyPhen [63] and SIFT [64]. Sequencing data were deposited in the European Genome-Phenome Archive (EGAS00001001915).

### 4.3. Identification of Mutational Driver Genes and Significantly Mutated Pathways

We employed two complementary approaches to detect positive selection signals in driver genes: OncodriveFM [65], implemented via the IntOGen platform [66], and MutSigCV [24]. Both algorithms were applied separately to mBCC and nodBCC samples. For OncodriveFM, a significance threshold was defined using a corrected *p*-value *q* < 0.1. Due to the small sample size, MutSigCV *p*-values could not be adjusted, so raw *p* < 0.05 was used, with genes having *p* < 0.02 highlighted in the results. Genes identified as significant by both methods and overlapping were selected for further analysis. Significantly mutated pathways were determined using OncodriveFM (*q* < 0.05), followed by comparative analysis between the mBCC and nodBCC groups.

### 4.4. Randomisation Comparison Test for Subtype Driver Genes

To pinpoint driver genes specific to mBCC or nodBCC, we focused on candidates that were significantly mutated in one subtype (*q* < 0.05) but not in the other (*q* > 0.1) based on OncodriveFM and MutSigCV results. A previously described randomisation procedure [67] was then applied, in which the complete set of 20 BCC samples was randomly divided into two equal groups of 10. The driver gene detection approaches described above were used within the split groups, and significance values for our subtype candidates were calculated. This randomisation was repeated 100 times, and an expected distribution of significance values was determined for each subtype driver candidate. Significant genes from the randomisation test (two-tail *p* < 0.05) were further noted and regarded as subtype-specific genes.

### 4.5. RNA Sequencing

RNA sequencing of tumour–stroma pairs for three patients in each mBCC and nodBCC subtype was performed. Extraction of high-quality RNA (RIN > 7 on Agilent RNA pico-chip) was required for RNA sequencing, which underwent 100 bp paired-end reads on an Illumina HiSeq 2000, averaging 13 million paired-end reads per sample. Output FASTQ files were aligned to the reference genome using TopHat2 [68]. Reads uniquely aligned to the exonic regions of each gene (mapping quality score Q > 10) were quantified using HTSeq [69] based on Ensembl annotation (Release 75). Only genes with at least one count per million (CPM) in a minimum of three samples were retained for analysis. Following scale normalisation, the read counts were transformed to log2(CPM) values using the voom function [70]. Analyses of differential gene expression between tumour and stroma samples were performed using the limma R package (version 3.48) [71], with the model of paired tumour–stroma comparison accounting for the baseline differences among patients. Gene-set enrichment analysis (GSEA) was performed for mBCC and nodBCC tumour–stroma comparisons to identify common and specific dysregulated pathways using the canonical pathways from the Molecular Signatures Database (MSigDB-C2 v5.0) [72]. Sequencing data were deposited into the Gene Expression Omnibus (GSE937997).

### 4.6. Real-Time Quantitative RT-PCR of Human Tissue and Keratinocyte Models

Total RNA was extracted using LCM as described above and reverse-transcribed to complementary DNA with SuperScript III First-Strand Synthesis SuperMix for qRT-PCR (Invitrogen, Waltham, MA, USA). Exon–exon junction-spanning primers were designed using Primer-BLAST (https://www.ncbi.nlm.nih.gov/tools/primer-blast/ accessed on 21 May 2021), and qRT-PCR reactions were performed in triplicate on an Applied Biosystems 7900HT Fast Real-Time PCR System using SYBR select master mix (Life Technologies). Relative transcript expression was calculated by the 2^−ΔΔCt^ method, and mRNA levels of each gene were normalised to the geometric mean of *GAPDH* and *β-actin* internal housekeeping genes. Statistical significance was determined using an unpaired, 2-tailed Student’s *t*-test. A *p*-value of less than 0.05 was considered significant. The primer sequences are in Appendix A.

### 4.7. PTCH1, FLNB, and PTCH1-FLNB Human Keratinocyte Knockdown BCC Models

NEB1 cells originate from early epidermal keratinocytes and were immortalised using HPV16 E6/E7 virus, as previously described [73]. NEB1 keratinocytes have previously been used by our group to generate a validated model of nBCC via stable PTCH knockdown in which both Hh pathway and sensitivity to Hh inhibitors have been studied [28]. PTCH1-knockdown keratinocytes were used to investigate the effects of FLNB knockout against a background null for PTCH that may better reflect BCC. NEB1 cells were cultured in keratinocyte growth medium (KGM) composed of Alpha MEM supplemented with 10% (*v*/*v*) heat-inactivated foetal bovine serum (Brazilian origin, Lonza, Basel, Switzerland), 2 mM L-glutamine, and 2% (*v*/*v*) penicillin–streptomycin (PAA Laboratories). The KGM supplement included 10 ng/mL epidermal growth factor (EGF), 0.5 µg/mL hydrocortisone, 5 µg/mL bovine insulin, 1.8 × 10^−4^ M adenine, and 1 × 10^−10^ M choleratoxin (Sigma, Kawasaki, Japan). Cells were incubated at 37 °C in a humidified atmosphere with 5% (*v*/*v*) CO_2_. Keratinocyte cell lines were retrovirally transduced with a PTCH1-targeting small-hairpin RNA (shRNA) construct specific for exon 24. A scrambled shRNA vector was used to generate a control cell line [28]. All sequences were previously cloned into the pSUPERIOR.retro.puro vector (Oligoengine, Seattle, WA, USA). Scramble control non-targeting sequence: GCGCGATATATAGAATACG. PTCH1 shRNA exon 24: AAGGAAGGATGTAAAGTGGTA.

### 4.8. Invasion Assays for the Organotypic Models

Matrigel and collagen-based organotypic cultures were established to assess cell invasion, as they provide an in vitro model that closely mimics skin. Primary fibroblasts were resuspended in 100 µL of RM+ medium and mixed with 200 µL of Matrigel (BD Pharmingen, NJ, USA) and 200 µL of type I collagen (Millipore, Watford, UK). The resulting mixture was gently poured into a 24-well plate and allowed to solidify in the incubator for approximately 30 min. Once the gel had set, NEB1 cells were seeded on top in 1 mL of RM+ medium and incubated overnight. The medium was subsequently replaced daily for 14 days, after which the gels were fixed in 4% formaldehyde, processed, and embedded in paraffin for tissue sectioning to evaluate the extent of cell invasion.

### 4.9. Immunohistochemistry and Immunofluorescence

Immunofluorescence (IF) was performed for semi-quantitative assessment and validation of Hh pathway expression in mBCC compared with nodBCC FFPE sections and was applied to our BCC models. Antigen retrieval was carried out using a DAKO EnVision™ FLEX + System (Dako, Carpinteria, CA, USA) at 97 °C for 20 min under alkaline (pH 9.0; PTCH1 1:1500, SMO 1:200, GLI1 1:250; Abcam (Cambridge, UK)) or acidic (pH 6.0; GLI2 1:150; Abcam) conditions. Sections were blocked with 5% goat serum for one hour, incubated with primary antibodies for 30 min, and then treated with an AlexaFluor-568 secondary antibody (Invitrogen). Nuclei were counterstained with DAPI (Sigma-Aldrich; 1:1000) to enable accurate cell quantification. Positive control tissues included breast, brain, testes, and intestine for PTCH1, SMO, GLI1, and GLI2, respectively. Negative controls, which omitted the primary antibody, were used to reduce background fluorescence. Statistical comparisons were conducted using Student’s *t*-test, with *p*-value < 0.05 considered significant. Figures present mean values, with error bars indicating SEM.

### 4.10. Confocal Microscopy and Fluorescence Signal Quantification

Immunofluorescence sections were examined using a Zeiss LSM710 Meta confocal laser microscope and a ZEN configuration tool (Carl Zeiss microscopy, Jena, Germany). Negative control slides were used to remove background fluorescence and set up the acquisition conditions that subsequently remained constant throughout the imaging process. Images were taken at 200 × magnification. Output TIFF images were analysed using ImageJ (http//imagej.nih.gov/ij; accessed on 26 October 2025; version 1.53) to quantify antibody expression. For semi-quantification of antibody expression, fluorescence intensity was determined in regions of interest (ROIs), ensuring a standardised area size whilst containing the same number of nuclei, as determined by the DAPI staining. Furthermore, 3 separate ROIs were taken for each area in each sample to obtain a mean signal for comparison. IF data are expressed as mean± SEM of a given number of observations. Comparison between groups was made using Student’s *t*-test. A *p*-value of less than 0.05 was considered to be significant. Figures show the mean, with error bars representing SEM.

### 4.11. Western Blotting for Antibody Validation

Validation of the aforementioned antibodies occurred against a known overexpressed Hh cancer cell line previously described and shown in Appendix A [74]. The technique was carried out as previously described using 1:500 anti-PTCH, 1:200 anti-SMO, 1:1000 anti-GLI1, and 1:500 anti-GLI2 (Abcam) [75].

## Figures and Tables

**Figure 1 ijms-26-11377-f001:**
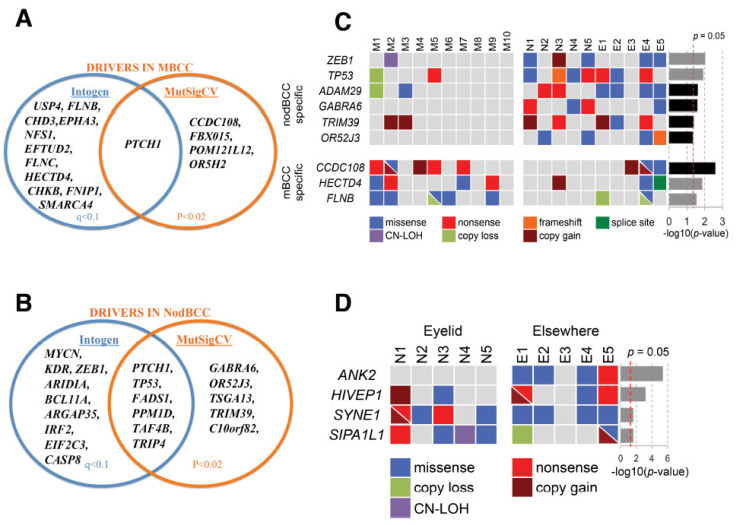
**Driver genes in basal cell carcinoma (BCC):** (**A**) driver genes for mBCC detected using OncodriveFM and MutSigCV; (**B**) driver genes for nodBCC detected using OncodriveFM and MutSigCV; (**C**) subtype-specific genes revealed by randomisation test differentiating mBCC and nodBCC. The grey and black *p*-value bars represent the randomisation test significance based on OncodriveFM and MutSigCV, respectively; (**D**) site-specific nodBCC genes revealed by randomisation test based on OncodriveFM statistics. *n* = 10 morphoeic BCC and 10 nodular BCC. Nod (N), periocular nodular BCC; M, periocular morphoeic BCC.

**Figure 2 ijms-26-11377-f002:**
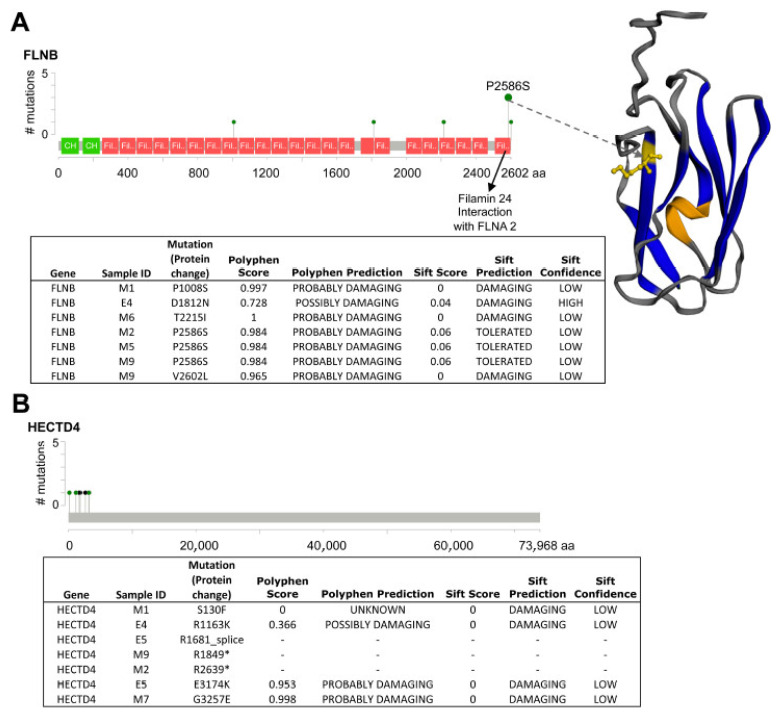
**Analysis of somatic variants in FLNB and HECTD4 protein domains.** (**A**) Somatic variants in FLNB. Variants seen in FLNB demonstrating a hotspot at P2856S in Filamin domain 24 of 3 BCC samples, which would potentially disrupt its interaction with FLNA2. A fourth variant, V2602L also sits within Filamin domain 24. The majority of FLNB mutations were predicted to be damaging by PolyPhen and SIFT. (**B**) Somatic variants in HECTD4. Variants seen in HECTD4 occurring at the N-terminal or start of the protein domain; however, little is known about its protein structure. Two stop codon gain mutations (*) and one mutation at the splice site were found, potentially disrupting its normal function.

**Figure 3 ijms-26-11377-f003:**
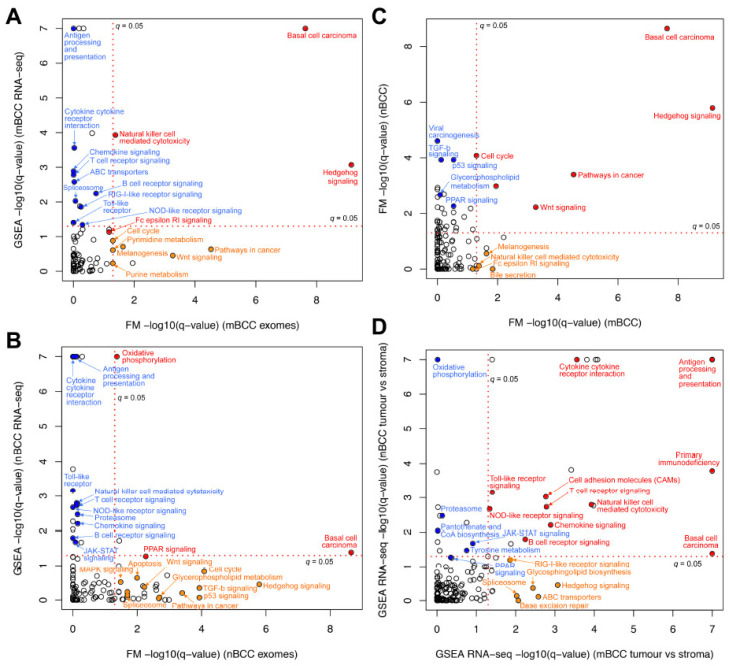
**Transcriptome and mutation pathway analysis and comparison within and between basal cell carcinoma subtypes.** (**A**) Morphoeic BCC transcriptome gene-set enrichment analysis (GSEA) compared to its WES OncodriveFM pathway data; (**B**) nodular BCC transcriptome GSEA compared to its WES OncodriveFM pathway data. Pathways in red are significantly changed in both DNA and RNA analysis using *q* = 0.05. Orange pathways represent pathways significantly altered in WES data only, and blue pathways are RNA-seq aberrantly expressed pathways only; (**C**) WES mutations of mBCC and nodBCC were compared using OncodriveFM pathway analysis. *n* = 10 morphoeic and 10 nodular BCCs; (**D**) transcriptome gene-set enrichment analysis (GSEA) of mBCC was compared to that of nodBCC. *n* = 3 morphoeic and 3 nodular BCCs. Red pathways are significantly changed in both histological subtypes using *q* = 0.05. Orange pathways represent pathways significantly altered in morphoeic BCC only, and blue pathways are nodular-specific pathways only.

**Figure 4 ijms-26-11377-f004:**
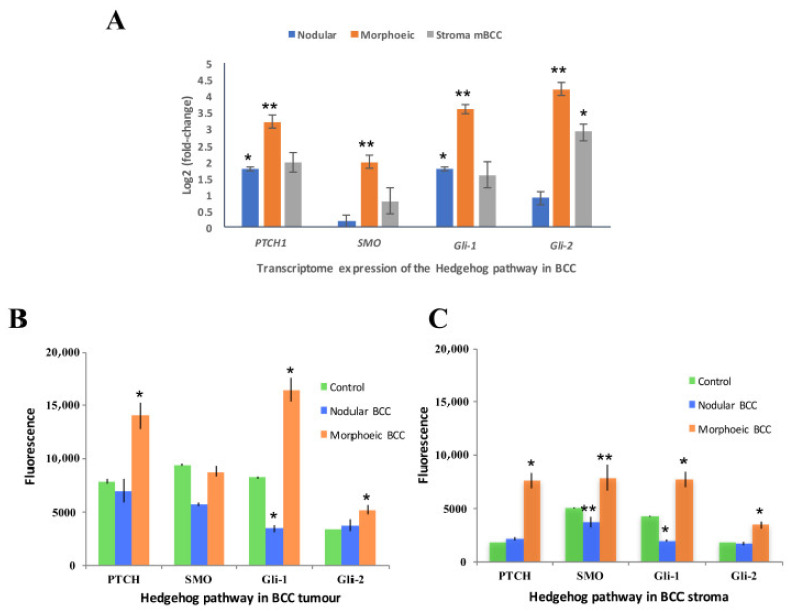
***GLI2* dominant expression of hedgehog (Hh) in morphoeic BCC including the stroma.** (**A**) qRT-PCR determined expression of Hh in morphoeic BCC, nodular BCC, and morphoeic stroma against normal eyelid. *n* = 3 morphoeic BCCs, 3 nodular BCCs, and 3 morphoeic stroma BCCs. Data presented as mean ± SEM; (**B**) immunofluorescence of Hh protein expression in BCC tumours. *n* = 10 morphoeic and 10 nodular BCC against physiologically active Hh control. Data presented as mean ± SEM; (**C**) immunofluorescence of Hh protein expression in BCC stroma. *n* = 10 morphoeic and 10 nodular BCCs against physiologically active Hh control. Data presented as mean ± SEM. * *p* < 0.05; ** *p* < 0.01.

**Figure 5 ijms-26-11377-f005:**
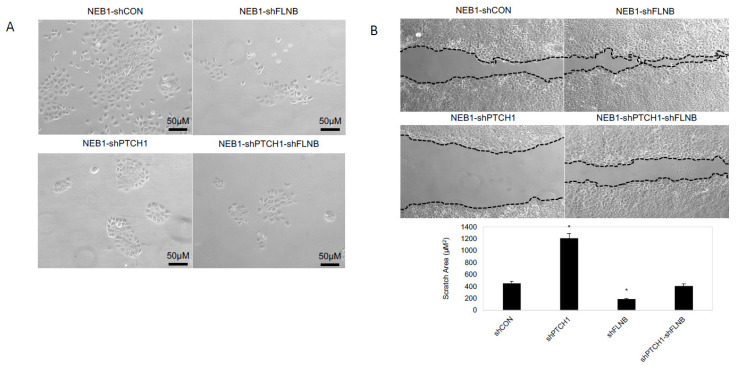
**FLNB and PTCH1-FLNB knockdown mimics morphoeic BCC phenotype.** (**A**) FLNB knockdown (NEB1-shFLNB) in NEB1 keratinocytes has little effect on keratinocyte morphology compared to control wildtype (NEB1) keratinocytes. PTCH knockdown (NEB1-shPTCH1) result in compact keratinocyte morphology, which resemble nodular BCC. Suppression of FLNB in PTCH knockdown cells (NEB1-shPTCH1-shFLNB) results in loss of this compact phenotype, and keratinocytes are more disorganised and do not form compact colonies. (**B**) In scratch assays NEB1-shPTCH1 migrate much less than control cells; however, FLNB suppression (NEB1-shPTCH1-shFLNB) enhances migration despite PTCH knockdown. Graph showing the area of the remaining space (µM^2^) 24 h post scratch. *, *p* ≤ 0.05 as calculated by the Student *t* test; error bars, SD.

**Figure 6 ijms-26-11377-f006:**
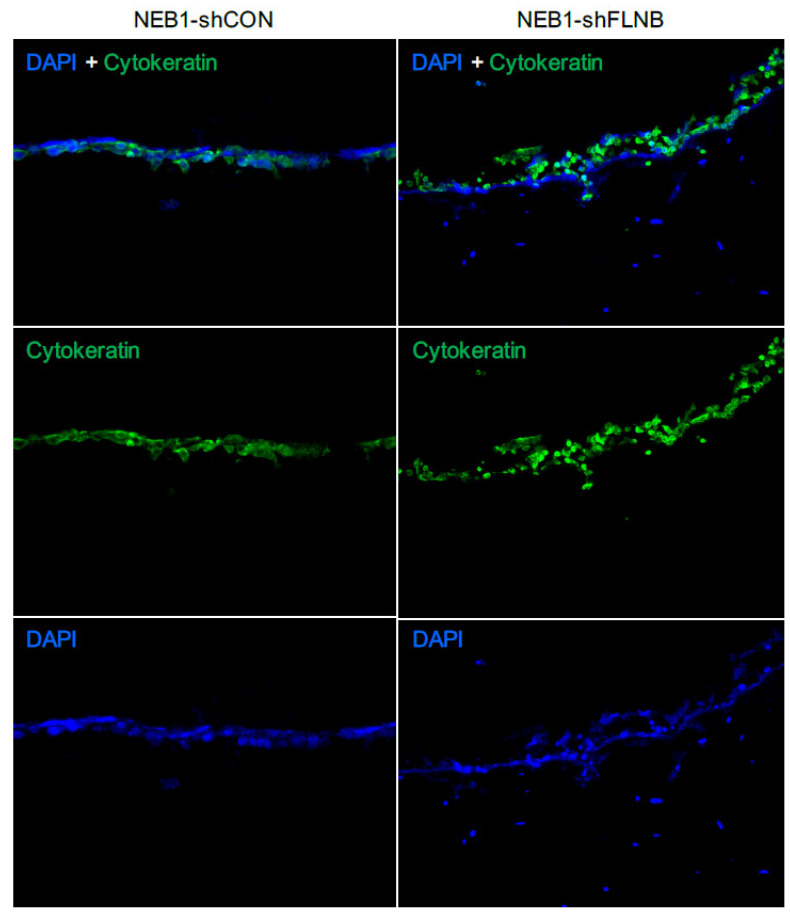
NEB1-shFLNB cells invade in 3D culture, mimicking a morphoeic phenotype: 3D collagen gel models of keratinocytes cultured with fibroblasts were stained for cytokeratin and DAPI to identify keratinocytes (double positive stained cells), while cytokeratin-negative DAPI cells are fibroblasts. The epidermal layer of NEB1-shFLNB cells showing a disorganised epithelial layer compared to shCON control with more invasion into the gel and apparent attraction of fibroblasts (DAPI only cells) in close proximity to the shFLNB cell layer. Invasion measured at 14 days; *n* = 3 for separate experiments.

**Figure 7 ijms-26-11377-f007:**
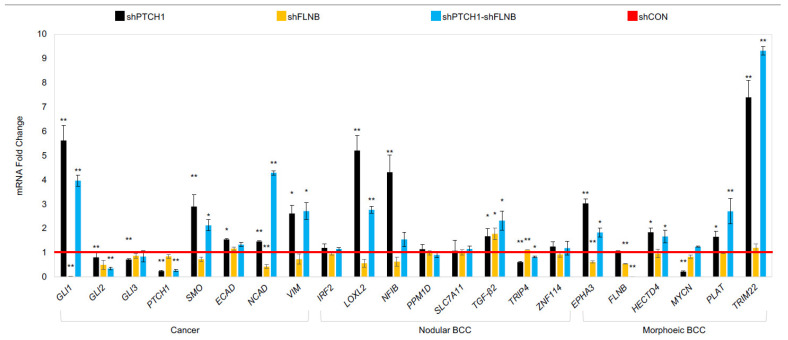
**Keratinocyte model gene expression of specific potential driver/suppressor genes that were identified in human BCC tissue.** Gene expression highlighted in NEB1-shPTCH, NEB1-shFLNB, NEB1-shFLNB-shPTCH compared to shCON control (red line). Data presented as the mean ± SEM for *n* = 3 separate experiments. * *p* ≤ 0.05, ** *p* ≤ 0.01.

## Data Availability

The datasets used and/or analysed during the current study are available in the Appendix A. All sequencing data were submitted to the relevant database: exome sequencing was deposited into the European Genome-Phenome Archive (EGAS00001001915) and RNA-sequencing was deposited into the Gene Expression Omnibus (GSE937997).

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
