# Peer review of "GLI2* and *FLNB* Define Periocular Morphoeic Basal Cell Carcinoma"

_ijms, 2025, doi:10.3390/ijms262311377_

Round 1
Reviewer 1 Report
Comments and Suggestions for Authors
The manuscript is interesting, provides valuable molecular insights into periocular morphoeic BCC and is overall well-written. I appreciated the clear presentation of data and the thoughtful design.
I have a few minor suggestuons:
- In the Results (Section 2.1, Figure 1a–c) and Methods (Section 4.3), the authors mention that several genes identified by OncodriveFM and MutSigCV (e.g. FLNB, HECTD4, and GLI2) were functionally validated. It would be useful if the authors could briefly explain why these particular genes were chosen for further in vitro experiments.
- While NEB1 keratinocytes are a reasonable model, they differ from periocular keratinocytes in their microenvironment. I suggest a short note on this limitation in the Discussion.
- The stromal WES and RNA-seq findings in Section 2.5 are interesting, particularly the shared somatic variants detected in both tumour and stromal compartments. However, since only one tumour–stroma pair was analysed by WES, these observations should be considered preliminary. I believe that conclusions regarding this finding should be formulated with caution and emphesize that these results are preliminary.
Author Response
|
Point-by-point response to Comments and Suggestions for Authors from Reviewer 1
|
|
Comments 1: In the Results (Section 2.1, Figure 1a–c) and Methods (Section 4.3), the authors mention that several genes identified by OncodriveFM and MutSigCV (e.g. FLNB, HECTD4, and GLI2) were functionally validated. It would be useful if the authors could briefly explain why these particular genes were chosen for further in vitro experiments.
|
|
Response 1: Thank you for pointing this out. Therefore, we have added a sentence to explain our choice and change can be found – page 9, line 193.
|
|
Comments 2: While NEB1 keratinocytes are a reasonable model, they differ from periocular keratinocytes in their microenvironment. I suggest a short note on this limitation in the Discussion.
Response 2: Thank you for pointing this out. We agree with this comment. Therefore, we have revised the manuscript and this change can be found – page 9, line 307.
|
|
Comments 3: The stromal WES and RNA-seq findings in Section 2.5 are interesting, particularly the shared somatic variants detected in both tumour and stromal compartments. However, since only one tumour–stroma pair was analysed by WES, these observations should be considered preliminary. I believe that conclusions regarding this finding should be formulated with caution and emphesize that these results are preliminary.
|
|
Response 3: Thank you for pointing this out. We agree with this comment. Therefore, we have revised the manuscript and these changes can be found –page 11, line 228 and page number 9, line 307.]
|
Reviewer 2 Report
Comments and Suggestions for Authors
Comments for Authors
This is a valuable and well-structured study addressing the molecular underpinnings of periocular morphoeic basal cell carcinoma (mBCC). The integration of whole-exome sequencing, transcriptomic analysis, and functional in-vitro modeling provides important insight into potential subtype-specific mechanisms.
However, several conceptual and interpretive issues require careful revision before publication.
Major Comments
Interpretation of FLNB Function
The statement that “FLNB is a potential tumour suppressor producing a morphoeic phenotype” is conceptually inconsistent.
If FLNB acts as a tumour suppressor, its loss—not its presence—should generate the morphoeic phenotype.
Please clarify the causal direction and revise the abstract and discussion accordingly (e.g., “Loss of FLNB produces a morphoeic phenotype, suggesting that FLNB functions as a tumour suppressor”).
Mechanistic Link Between GLI2 and Morphoeic Behaviour
While GLI2 overexpression is clearly observed in mBCC, the current data only demonstrate correlation, not causation.
The phrasing “GLI2 overexpression defines the morphoeic phenotype” should be moderated to “GLI2 overexpression characterises or is associated with the morphoeic phenotype”, unless direct functional assays (knockdown, inhibition, rescue) are added.
Subtype-Specificity Claims
The conclusion that FLNB and HECTD4 are mBCC-specific drivers is based on a relatively small cohort (n=10).
The authors should provide additional statistical support (e.g., permutation or bootstrapping) or use more cautious language such as “putative morphoeic-specific modifiers” rather than “drivers.”
Definition of the “Morphoeic-like” Phenotype in Vitro
The description of FLNB or PTCH1-FLNB knockdown as producing a “morphoeic-like phenotype” requires clarification.
What specific morphological or functional parameters (colony disorganisation, migration index, invasion, ECM remodeling) were used to define this phenotype?
Quantitative metrics should be added or clearly referenced in the figure legends.
Over-Interpretation of “Dominant GLI2 Signalling”
The claim that “WES, GSEA and protein validation demonstrate GLI2-dominated aberrant Hh expression” seems overstated.
Without quantitative comparison to GLI1 expression levels or functional validation, the term “dominant” should be replaced by “predominant pattern of GLI2 activation.”
Speculative Statements on Non-canonical GLI2 Regulation
The discussion proposes a “signal-induced regulatory mechanism outside canonical Hh signalling”, possibly involving PDGFB.
This is an interesting hypothesis but lacks experimental evidence in the current study.
It should be clearly presented as a speculation or future direction, not as a demonstrated mechanism.
Clinical Correlation and Biological Plausibility
The final paragraph links recurrence risk to “failure to remove inflammatory material.”
This explanation is surgical rather than molecular and does not align with the transcriptomic focus of the paper.
Consider rephrasing to emphasize persistent stromal inflammation, EMT-associated crosstalk, or tumour–stroma interaction as biologically plausible causes of recurrence.
Minor Comments
The introduction focuses heavily on histological description. It would benefit from a brief rationale explaining why the eyelid microenvironment (intermittent UV exposure, fibroblast composition) may predispose to distinct genetic drivers.
The RNA-seq results suggest downregulation of FLNB and HECTD4, but expression loss alone cannot prove loss-of-function mutation. Please phrase cautiously.
Some figures (especially those showing knockdown phenotypes) would benefit from quantitative data or representative images with scale bars.
Consider including a summary schematic showing how FLNB loss and GLI2 upregulation interact within the Hedgehog pathway to produce the invasive phenotype.
The sample size (10 mBCC, 10 nodBCC) should be explicitly stated in the Methods and acknowledged as a limitation.
Overall Assessment
This study offers novel and potentially important findings linking FLNB loss and GLI2 upregulation to the pathogenesis of periocular morphoeic BCC.
However, several mechanistic interpretations currently exceed the strength of the presented data.
Clarifying these causal relationships, refining terminology, and tempering speculative conclusions will greatly strengthen the manuscript.
Recommendation: Major Revision.
Author Response
Point-by-point response to Comments and Suggestions for Authors from Reviewer 2
Comments 1: Interpretation of FLNB Function
The statement that “FLNB is a potential tumour suppressor producing a morphoeic phenotype” is conceptually inconsistent.
If FLNB acts as a tumour suppressor, its loss—not its presence—should generate the morphoeic phenotype.
Please clarify the causal direction and revise the abstract and discussion accordingly (e.g., “Loss of FLNB produces a morphoeic phenotype, suggesting that FLNB functions as a tumour suppressor”).
Response 1:Thank you for pointing this out. We agree with this comment. Therefore, we have revised the manuscript these changes can be found – Page 1, line 32 updated the abstract; page number 11, line 238 updated the discussion; page 12, line 313 updated the conclusion
Comments 2: Mechanistic Link Between GLI2 and Morphoeic Behaviour
While GLI2 overexpression is clearly observed in mBCC, the current data only demonstrate correlation, not causation.
The phrasing “GLI2 overexpression defines the morphoeic phenotype” should be moderated to “GLI2 overexpression characterises or is associated with the morphoeic phenotype”, unless direct functional assays (knockdown, inhibition, rescue) are added.
Response 2: Thank you for pointing this out. We agree with this comment. Therefore, we have revised the manuscript, and this change can be found – page number 6, line 151.
Comments 3: Subtype-Specificity Claims
The conclusion that FLNB and HECTD4 are mBCC-specific drivers is based on a relatively small cohort (n=10).
The authors should provide additional statistical support (e.g., permutation or bootstrapping) or use more cautious language such as “putative morphoeic-specific modifiers” rather than “drivers.”
Response 3: Thank you for pointing this out. We agree with this comment. Therefore, we have revised the manuscript, and this change can be found – page number 12, line 313.
Comments 4: Definition of the “Morphoeic-like” Phenotype in Vitro
The description of FLNB or PTCH1-FLNB knockdown as producing a “morphoeic-like phenotype” requires clarification.
What specific morphological or functional parameters (colony disorganisation, migration index, invasion, ECM remodeling) were used to define this phenotype?
Quantitative metrics should be added or clearly referenced in the figure legends.
Response 4: Thank you for pointing this out. We agree with you comment regarding the need for clarification. We used colony disorganisation, enhanced migration in the scratch assay and invasion in the 3D organotypics. There are quantitative metrics in Figure 5. We have made this more explicit in the results and methods section: Page 9, line 202 and page 15 line 427-436
Comments 5: Over-Interpretation of “Dominant GLI2 Signalling”
The claim that “WES, GSEA and protein validation demonstrate GLI2-dominated aberrant Hh expression” seems overstated.
Without quantitative comparison to GLI1 expression levels or functional validation, the term “dominant” should be replaced by “predominant pattern of GLI2 activation.”
Response 5: Thank you for pointing this out. We agree with this comment. Therefore, we have revised the manuscript, and this change can be found – page number 12, line 282.
Comments 6: Speculative Statements on Non-canonical GLI2 Regulation
The discussion proposes a “signal-induced regulatory mechanism outside canonical Hh signalling”, possibly involving PDGFB.
This is an interesting hypothesis but lacks experimental evidence in the current study.
It should be clearly presented as a speculation or future direction, not as a demonstrated mechanism.
Response 6: Thank you for pointing this out. We agree with this comment. Therefore, we have revised the manuscript, and this change can be found – page number 12, line 292.
Comments 7: Clinical Correlation and Biological Plausibility
The final paragraph links recurrence risk to “failure to remove inflammatory material.”
This explanation is surgical rather than molecular and does not align with the transcriptomic focus of the paper.
Consider rephrasing to emphasize persistent stromal inflammation, EMT-associated crosstalk, or tumour–stroma interaction as biologically plausible causes of recurrence.
Response 7: Thank you for pointing this out. We agree with this comment. Therefore, we have revised the manuscript, and this change can be found – page number 13, line 324.
Comments 8: The introduction focuses heavily on histological description. It would benefit from a brief rationale explaining why the eyelid microenvironment (intermittent UV exposure, fibroblast composition) may predispose to distinct genetic drivers.
Response 8: Thank you for pointing this out. We agree with this comment. Therefore, we have revised the manuscript, and this change can be found – page number 2, line 48.
Comments 9: The RNA-seq results suggest downregulation of FLNB and HECTD4, but expression loss alone cannot prove loss-of-function mutation. Please phrase cautiously.
Response 9: Thank you for pointing this out. We agree with this comment. Therefore, we have revised the manuscript and this change can be found – page number 3, line 104.
Comments 10: Some figures (especially those showing knockdown phenotypes) would benefit from quantitative data or representative images with scale bars.
Response 10: Thank you for pointing this out. The quantitative data for the NEB1 cells including knockdown phenotypes are shown in Figure 6. We have made this clear on page 9. line 209
Comments 11: Consider including a summary schematic showing how FLNB loss and GLI2 upregulation interact within the Hedgehog pathway to produce the invasive phenotype.
Response 11: Thank you for pointing this out. As you rightly mentioned in comment 6, I do not think we can create a schematic as it is likely to be a non-conical route which as yet, we do not know the correct interaction and any schematic would be speculation.
Comments 12: The sample size (10 mBCC, 10 nodBCC) should be explicitly stated in the Methods and acknowledged as a limitation.
Response 12: Thank you for pointing this out. We agree with this comment. Therefore, we have revised the manuscript, and this change can be found – Methods section page number 13, line 331. Acknowledged as a limitation in the discussion on page number 12, line 317